# Protective Role of IRBIT on Sodium Bicarbonate Cotransporter-n1 for Migratory Cancer Cells

**DOI:** 10.3390/pharmaceutics12090816

**Published:** 2020-08-27

**Authors:** Soyoung Hwang, Dong Min Shin, Jeong Hee Hong

**Affiliations:** 1Department of Physiology, College of Medicine, Department of Health Sciences and Technology, GAIHST, Gachon University, 155 Getbeolro, Yeonsu-gu, Incheon 21999, Korea; snrntlwy1004@gmail.com; 2Department of Oral Biology, Yonsei University College of Dentistry, Seoul 03722, Korea

**Keywords:** IRBIT, migration, sodium bicarbonate cotransporter, cancer metastasis

## Abstract

IP_3_ receptor-binding protein released with IP_3_ (IRBIT) interacts with various ion channels and transporters. An electroneutral type of sodium bicarbonate cotransporter, NBCn1, participates in cell migration, and its enhanced expression is related to cancer metastasis. The effect of IRBIT on NBCn1 and its relation to cancer cell migration remain obscure. We therefore aimed to determine the effect of IRBIT on NBCn1 and the regulation of cancer cell migration due to IRBIT-induced alterations in NBCn1 activity. Overexpression of IRBIT enhanced cancer cell migration and NBC activity. Knockdown of IRBIT or NBCn1 and treatment with an NBC-specific inhibitor, S0859, attenuated cell migration. Stimulation with oncogenic epidermal growth factor enhanced the expression of NBCn1 and migration of cancer cells by recruiting IRBIT. The recruited IRBIT stably maintained the expression of the NBCn1 transporter machinery in the plasma membrane. Combined inhibition of IRBIT and NBCn1 dramatically inhibited the migration of cancer cells. Combined modulation of IRBIT and NBCn1 offers an effective strategy for attenuating cancer metastasis.

## 1. Introduction

The inositol 1, 4, 5-trisphosphate (IP_3_) receptor-binding protein released with IP_3_ (IRBIT, also referred to as AHCYL1) is a binding protein of IP_3_ receptor (IP_3_R) and mediates signaling events with various receptors [1,2,3]. The released IRBIT from IP_3_R plays a role in regulating bicarbonate secretion by interacting with and activating ion transporters, including cystic fibrosis transmembrane conductance regulator (CFTR), sodium bicarbonate cotransporters NBCe1-B and NBCn1, sodium hydrogen exchanger 3 (NHE3), chloride bicarbonate exchanger SLC26A6, and protein kinases, including calcium/calmodulin-dependent protein kinase IIα (CaMKIIα) [2,4,5,6]. Moreover, the IRBIT suppresses the activity of the IP_3_ receptor by competing with IP_3_ for the common binding site on the IP_3_ receptor [7]. The structure of IRBIT has been reported to possess an AHCY domain, coiled-coil domain, post synaptic density 95/drosophila disc large tumor suppressor/zonula occludens-1 (PDZ) domain, and N-terminal serine-rich region containing multiple phosphorylation sites [8,9]. IRBIT knockout mice with ovarian cancer have behavioral abnormalities and defects in the secretory function of the pancreatic ducts, suggesting that IRBIT has multiple in vivo functions [3,10].

Intracellular and extracellular pH homoeostasis are essential features of all cells, including cancer cells. The metabolism of cancer cells is an essential process that produces acidic metabolic products, including proton, lactate, and carbon dioxide [11]. The extracellular acidic pH affects proliferation, migration, invasion, metastasis, and therapeutic response of cancer cells via transmembrane-associated ion transporters and exchangers [12,13,14,15,16]. The electroneutral sodium bicarbonate cotransporter 1 (NBCn1) is a member of the solute carrier 4 (SLC4) family of bicarbonate transporter proteins and a major mediator of net cellular acid extrusion in most tissues [17,18,19]. The NBCn1 is expressed in numerous cells and tissues, including the kidneys, heart, gastrointestinal tract, muscles, testes, and brain [20,21,22,23]. NBCn1 mediates the uptake of bicarbonate ions in vascular smooth muscle cells and promotes cell migration and proliferation [24,25]. It has been also reported that NBCn1 expression is related to breast cancer and is increased in primary breast carcinomas and metastasis compared to normal breast tissue [26]. Moreover, NBCn1 promotes acid extrusion from breast cancer cells, maintains an alkaline intracellular microenvironment, and facilitates aggressive tumor development [27]. The cellular existence and development are associated with various ion channels and transporters by maintaining factors such as cell volume, membrane potential, and acid–base balance [28]. As a regulatory molecule, IRBIT binds and regulates the activity of ion channels and transporters such as CFTR [2], NHE3 [29], and SLC26A6 [30]. IRBIT also stimulates the activity of NBCs [2,30,31]. Although various binding partners of IRBIT have been identified, the physiological relevance of the interactions between IRBIT and NBCn1 in cancer cell migration remains to be elucidated. In addition, the information of role of IRBIT on tumor cells is relatively low. Thus, there is lots of potential in cancer biology within the scope of transporter physiology and involvement of regulatory proteins such as IRBIT in migratory machinery. We therefore aimed to investigate the role of IRBIT in regulating cancer cell migration via modulation of NBCn1 activity. We also verified the role of IRBIT in cancer cell migration by stimulation with oncogenic epidermal growth factor (EGF).

## 2. Materials and Methods

### 2.1. Plasmids and Cell Culture

Generation of the NBCn1 construct in pcDNA3.1 has been previously described [2]. cDNAs encoding the full-length IRBIT and GFP-IRBIT proteins were subcloned into pcDNA3.0 and pEGFP-C1, respectively. A549 lung adenocarcinoma cell line and HEK293T cells were obtained from the American Type Culture Collection (Rockville, MD, USA) and maintained in Dulbecco’s modified Eagle’s medium (DMEM, 11995073, Thermo Fisher Scientific, Waltham, MA, USA). MCF-7 cells were obtained from the American Type Culture Collection (Rockville, MD, USA) and maintained in mammalian cell culture medium (RPMI, A1049101, Thermo) supplemented with 10% fetal bovine serum (FBS, 1600-044, Invitrogen, Carlsbad, CA, USA) and 100 U/mL penicillin-streptomycin (15140122, Invitrogen), and incubated at 37 °C in a humidified incubator, in an atmosphere of 5% CO_2_ and 95% air. When the cells reached 80% confluence, the culture medium was removed, and the cells were washed with Dulbecco’s phosphate-buffered saline (DPBS, LB001-02, Welgene, Gyeongsan, Korea), followed by treatment with trypsin/ethylenediaminetetraacetic acid (EDTA) for 5 min. The dispersed cells were either transferred to fresh culture dishes for Western blotting, agarose spot assay, transwell assay, and MTT assay, or transferred to fresh culture dishes with glass coverslips for imaging and immunofluorescence experiments.

### 2.2. Agarose Spot Assay for Cell Invasion

Directional cell migration was examined by performing an agarose spot assay as previously described [32,33], with modifications, in the chemotactic invasion assay protocol. Briefly, 10 mg of agarose (UltraKem LE, Young Sciences, Bucheon, Korea) was placed in a 50 mL conical tube and diluted with 2 mL of physiological salt solution A (PSA; (numbers indicate mM) 140 sodium chloride (NaCl), 10  *_D_*-glucose, 5 potassium chloride (KCl), 1 magnesium chloride (MgCl_2_), 1 calcium chloride (CaCl_2_), 10  HEPES (pH 7.4)) for preparing a 0.5% agarose solution, which was spotted onto 6-well plates (Thermo) with four agarose spots per plate, and allowed to cool for 8 min at 4 °C. The cells were then plated at a density of 4 × 10^5^ cells/plate and allowed to adhere for 4 h before replacing with a medium supplemented with 0.1% FBS (Invitrogen) and 100 unit/mL penicillin (Invitrogen) in DMEM. The cell images were obtained after 4, 24, 48, and 72 h of incubation at 37 °C using the Meta Morph software (Molecular Devices, San Jose, CA, USA) with a 10× objective (Olympus, Tokyo, Japan). The cells observed beneath the agarose spots represented the migrated cells and were counted.

### 2.3. MTT Assay for Cell Viability

A549 cells were cultured in 96-well plates at a density of 5 × 10^3^ cells/well and were treated with S0859 (18497, Cayman, Ann Arbor, MI). After the indicated incubation period, the medium was replaced with MTT (3-(4, 5-dimethyl thiazol-2-yl)-2, 5-diphenyltetrazolium bromide, 2 mg/mL, MT1036, BioPrince, Chuncheon, Korea) solution and the cells were incubated for 2 h at 37 °C. The formazan crystals were dissolved by replacing with dimethyl sulfoxide (DMSO). The absorbance was subsequently determined at 570 nm with a UVM 340 microplate reader (VICTOR X3, PerkinElmer, Waltham, MA, USA).

### 2.4. Measurement of NBC Activity

The cells were attached onto coverslips and loaded with 6 μM 2′, 7′-bis-(carboxyethyl)-5-(and-6)-carboxyfluorescein (BCECF-AM, 0061, TEFlabs Inc, Austin, TX, USA) along with the same volume of 0.05% pluronic acid (P-3000MP, Invitrogen) for 15 min at room temperature (RT). After incubation with the BCECF dye, the cells were perfused with physiological salt solution, as previously described [34], for at least 5 min prior to determining the intracellular pH (pH_i_). pH_i_ was determined by measuring BCECF fluorescence using dual excitation wavelengths of 495 and 440 nm, and an emission wavelength of 530 nm. NBC activity was measured by incubating the cells with a CO_2_-saturated HCO_3_^—^ buffered ((numbers indicate mM) 120 NaCl, 10 *D*-glucose, 5 KCl, 1 MgCl_2_, 1 CaCl_2_, 2.5 HEPES (pH 7.8]) 25 sodium bicarbonate (NaHCO_3_)) medium containing 5-(N-ethyl-N-isopropyl)-amiloride (EIPA, 1154-25-2, Sigma Aldrich, Saint-Louis, MO, USA), followed by acidification with an Na^+^-free HCO_3_^−^-buffered medium. Images were obtained with a CCD camera (Retiga 6000, Q-Imaging) attached to an inverted microscope (Olympus) and analyzed with a Meta Fluor system (Molecular Devices). The images were individually normalized by subtracting the background fluorescence from the raw background signals.

### 2.5. Treatment with Small Interfering RNA (siRNA) and DNA Transfection

siRNA for human IRBIT was produced using Double-Promoter pFIV-H1/U6 siRNA cloning and expression vectors (SI111A-1, System Biosciences, Palo Alto, CA, USA), according to the instructions provided with the kit. The purified plasmids contained the human siRNA-IRBIT sequence (sense: 5′-AAA GGG CCA TGA ACG TCA ATG ATT CTG TTA C-3′; and antisense: 5′-AAA AGT AAC AGA ATC ATT GAC GTT CAT GGC C-3′) and siRNA-NBCn1 sequence (sense: 5′-AAA GGT GGG ATC CTT CTA TAC GCA TAG AA-3′; and antisense: 5′-AAA ATT CTA TGC GTA TAG AAG GAT CCC ACC TT-3′). A549 cells were transfected with 1 μg of the siRNA vectors. The siRNA vectors were diluted with 200 μL of Opti-Eagle’s minimum essential medium (Opti-MEM^TM^, 31985-070, Invitrogen) and mixed with Lipofectamine 2000 DNA transfection reagent. The mixture was incubated at RT for 25 min, transferred to cell dishes containing culture media, and cultured for 4 h. The transfected medium was replaced with fresh culture medium supplemented with FBS, and the cells were used after 48 h of transfection. The plasmid DNA was transfected with Lipofectamine 2000 according to the manufacturer’s protocol (11668019, Invitrogen). The plasmid DNA was individually diluted with 200 μL of Opti-MEM, to which 4 μL of Lipofectamine 2000 was added, and the mixture was incubated for 5 min at RT with 200 μL of the same medium. After 25 min, the DNA samples and Lipofectamine 2000 were mixed and added to the cell culture dish containing glass coverslips. Following 4 h of incubation, the medium was replaced with fresh DMEM supplemented with 10% FBS, and the cells were cultured for 24 h (for DNA transfection) or 48 h (for siRNA transfection) after transfection.

### 2.6. Immunofluorescence and Confocal Imaging

A549 cells were transferred onto cover glasses and fixed with chilled (−20 °C) methanol. The fixed cells were treated with 5% goat serum (in DPBS) for 1 h at RT for blocking the non-specific binding sites. The cells were incubated overnight with primary antibodies at 4 °C, followed by washing thrice with the incubation buffer (5% BSA in DPBS). The NBCn1 (ab82335, Abcam, Cambridge, UK) antibody was detected by treating the cells with secondary antibodies, rhodamine-tagged goat immunoglobulin G (IgG) (Jackson ImmunoResearch, West Grove, PA, USA, anti-mouse: 115-025-072, anti-rabbit: 111-025-144), and fluorescein isothiocyanate (FITC, Jackson ImmunoResearch, anti-mouse: 115-095-071, anti-rabbit: 111-095-003) for 1 h at RT. Following incubation, the cells were washed thrice with DPBS, and the cover glasses were mounted on glass slides using 20 μL of Fluoromount-G^TM^ containing 4′,6-diamidino-2-phenylindole (DAPI, 17984-24, Electron Microscopy Sciences, Hatfield, PA, USA), and incubated overnight at 4 °C. The slides were then analyzed using an LSM 700 Zeiss confocal microscope (Carl Zeiss, Germany) with ZEN software (Carl Zeiss, Oberkochen, Germany).

### 2.7. Surface Biotinylation, Co-Immunoprecipitation, and Western Blotting

The cells that had been transfected with siRNA and treated with S0859 (18497, Cayman) were incubated with 1× lysis buffer (9803, Cell signaling, Danvers, MA, USA) containing 20 mM Tris, 150 mM NaCl, 2 mM EDTA, 1% Triton X-100, and a protease inhibitor mixture that had been prepared by passing cell lysates after sonication. The cells were centrifuged at 11,000× *g* for 15 min at 4 °C, and the protein concentration was determined by Bradford assay (5000001, Bio-Rad, Hercules, CA, USA). For co-immunoprecipitation (Co-IP), the supernatant was treated with 1 μg/mL of the indicated antibodies at 4 °C for 16 h with gentle shaking, followed by incubation with agarose G protein beads (Santa Cruz, Dallas, TX, USA) for 4 h. The mixture was subsequently centrifuged at 11,000× *g* for 2 min at 4 °C and washed twice with the lysis buffer at 4 °C for 10 min. The beads were incubated in the sample buffer at 37 °C for 15 min for detaching the proteins. The eluted proteins were analyzed by Western blotting. To demonstrate the surface expression of the proteins, the transfected cells were incubated with 0.5 mg/mL EZ-LINK sulfo-NHS-LC-biotin (21335, Thermo) for 30 min on ice, followed by treatment with 100 mM of cold glycine solution for 10 min. The incubated cells were washed with DPBS and incubated with the lysis buffer. The cell extracts were centrifuged at 11,000× *g* for 15 min at 4 °C, and the cellular debris was discarded. The supernatants were incubated overnight with 80 μL Avidin beads (20347, Thermo) at 4 °C, followed by washing the beads with the lysis buffer. The collected beads were incubated with a protein sample buffer at 37 °C for 15 min for recovering the proteins. The warmed protein samples (30 μg) were separated by sodium dodecyl sulfate polyacrylamide gel electrophoresis (SDS-PAGE) and transferred onto polyvinylidene difluoride (PVDF, 1620177, Bio-rad) membranes soaked in methanol. The membrane was blocked with 5% non-fat milk solution in TBS-T (Tris-buffered saline (TBS) and 0.5% Tween-20) for 1 h. The membrane was subsequently incubated overnight with β-actin (A5441, Sigma, Saint-Louis), IRBIT (10658-3, Proteintech Group Inc, Rosemont, IL, USA), NBCn1 (ab82335, Abcam), GFP (ab6556, Abcam), and HA-tag (C29F4, Cell signaling) antibodies at 4 °C, and washed thrice with TBS-T. After washing, the membranes were incubated with horseradish peroxidase (HRP)-conjugated anti-mouse and anti-rabbit secondary antibodies, and the protein bands were visualized using an enhanced luminescent solution (32209, Thermo).

### 2.8. Transwell Membrane Immunostaining

Directional cell migration was examined by performing a transwell membrane immunostaining with Boyden chamber as previously described [35,36]. A549 cells (200 μL, 5 × 10^4^ cells) were cultured in each well of the upper chamber of 6-well plate. The bottom chambers were filled with pH 7.4 media, S0859, or EGF along with 1% FBS added to DMEM (500 μL). After incubation for 6 h, the membrane was subsequently stained with DAPI or crystal violet. Briefly, chilled methanol (−20 °C) was added to the plates and the cells were incubated for 1 min. The methanol was removed, and the cells were washed with DPBS. DAPI solution, mixed with distilled water (DW), was added to the plates, and the cells were incubated for 30 min in the dark. The media were carefully removed from the top and bottom plates. DW was added to the plates at RT and measured at 340 nm using an LSM 700 confocal laser scanning microscope (Carl Zeiss, Germany). Migration of the A549 cells after 6 h was determined by evaluating the number of nuclei that were stained with DAPI on the transwell membrane. For crystal violet staining, 0.25% crystal violet was added to the plate and the membrane was incubated for 15 min at RT. The crystal violet was subsequently removed, and the membrane was washed with DPBS. The medium was carefully removed from the top and bottom plates, following which DW was added to the plate and incubated for 30 min at RT. The plate was subsequently analyzed using an LSM 700 Zeiss confocal microscope (Carl Zeiss) with Mosaic software (Opto Science, Tokyo, Japan).

### 2.9. Scratch Wound Healing Assay

Scratch wound healing assay was performed according to previously described [37,38]. A549 cells (5 × 10^4^ cells) were cultured in 6-well plate. When the cell confluence reached about 80% and above and scratch wounds were produced with 1000 μL pipet tips in each well. After scratching, the cell debris was removed and the cell images were obtained after 0, 24, 48, and 72 h of incubation at 37 °C, using the 700 Zeiss confocal microscope (Carl Zeiss) with Mosaic software (Opto Science).

### 2.10. Statistical Analyses

All data from the indicated number of experiments were expressed as the mean ± standard error of the mean (SEM). The statistical differences between mean values obtained from the two or more sample groups were analyzed using paired Student’s *t*-test. Two independent sample datasets come from distributions with different of two different groups. Statistical significance was determined by analysis of variance for each experiment (* *p* < 0.05, ** *p* < 0.01, *** *p* < 0.001).

## 3. Results

### 3.1. IRBIT Enhanced Cell Motility via NBC

In order to demonstrate the role of IRBIT in cancer cell invasion and motility, we modified the agarose spot migration assay as previously described in our benchmark study [32]. Exposure of cancer cells to the microenvironment is known to increase tissue rigidity during tumor development and progression [39,40]. This assay provides a more rigid environment than that of the culture media and effectively mimics the microenvironment of cancer cells, allowing the function of cellular invadopodia to be studied more precisely. The migration ability of cells in the agarose spot was determined. The experimental scheme is presented in Figure 1A,B. In order to verify the role of NBC in the migratory function of cancer cells, the activity of NBC was measured in region of interests of two regions, namely, the migration area and steady area. The activity of NBC in the migration area was found to be 2-fold higher than that in the steady area (Figure 1C,D). To explore the role of IRBIT on cell migration, the motility of the invasive A549 cells in the agarose spots was compared between the control and IRBIT-overexpressed conditions (Figure 1E). It was observed that, on average, the invasive motility of IRBIT-expressing A549 cells was approximately 1.4-fold higher than that of the control cells (Figure 1F). We also confirmed the migration assay with DAPI and crystal violet to verify the efficacy of IRBIT plasmids (Appendix A) and enhancement of migration of IRBIT by scratch wound healing assay (Appendix A). IRBIT recruits NBCe1-B and enhances the activity of NBC [31]. To explore the role of the transporter in cell motility, NBC imaging was performed. The role of IRBIT in regulating NBC activity was investigated by studying the activity of NBC in IRBIT-overexpressing A549 cells. NBC activity was enhanced in the IRBIT-overexpressing cells (Figure 1G,H). There was no difference between the two groups with respect to the resting pH_i_ level (Figure 1I). These results suggested that IRBIT enhanced cell motility by increasing NBC activity.

### 3.2. IRBIT Co-Localized with NBCn1 and Modulated the Membrane Expression of NBCn1

NBCn1 is closely associated with the cancer microenvironment and is necessary during development of cancer [27]. To investigate the protein–protein interactions and colocalization between IRBIT and NBCn1, we determined the effect of IRBIT on the interaction and immunolocalization of NBCn1 in HEK293T cells. As depicted in Figure 2A, Co-IP data show the interaction between IRBIT and NBCn1. Additionally, A549 cells were separately immunostained with NBCn1 and IRBIT. In order to ensure that there were no artifacts from the same host during co-localization of the two proteins, the cells were overexpressed GFP-IRBIT. It was observed that NBCn1 localized in the plasma membrane, and IRBIT co-localized with NBCn1 (Figure 2B,C). The membrane localization and surface expression of NBCn1 were enhanced by IRBIT (Figure 2D–F). In order to confirm that IRBIT enhanced the activity of NBCn1, HEK293T cells were transfected with NBCn1. The IRBIT increased NBC activity in the cells that overexpressed NBCn1 (Appendix A). The results suggest that IRBIT enhanced cell motility by increasing the activity of NBCn1.

### 3.3. IRBIT Knockdown Decreased Cell Motility and NBC Activity

To verify the role of IRBIT on NBC activity, cells were transfected with SiRNA-IRBIT. The efficacy of SiRNA-IRBIT was evaluated by Western blotting (Figure 3A,B). The invasive motility of the cancer cells was inhibited by IRBIT knockdown (Figure 3C,D). IRBIT knockdown dramatically inhibited the activity of NBC (Figure 3E,F). The resting pH value of cells with SiRNA-IRBIT was lower than that of the control (Figure 3G). The surface expression (Figure 3H) and membrane localization (Figure 3I,J) of NBCn1 decreased in the presence of SiRNA-IRBIT. IRBIT knockdown reduced the activity of NBC and the membrane localization of NBCn1, suggesting that IRBIT is involved in stabilizing NBCn1 on the plasma membrane for cell migration.

### 3.4. The NBC Inhibitor, S0859, Partially Inhibited Cell Motility

In order to confirm the role of NBC in cell migration, cells were treated with the NBC inhibitor S0859. It was observed that cell motility following S0859 treatment was approximately 50% lower than that of the control (Figure 4A,B). However, S0859 did not affect cell viability (Figure 4C). The membrane localization and surface expression of NBCn1 did not change in the presence of S0859 (Figure 4D–F). We also confirmed that the activity of NBC was inhibited by S0859 (Figure 4G,H). An acidic environment is a characteristic feature of cancerous tissues [41,42,43]. The inhibitory role of S0859 on the migration of cancer cells to an acidic environment was confirmed by the transwell assay, performed at normal physiological pH and at acidic pH. The cells migrated toward the bottom chamber, which was at acidic pH. Cell migration was determined by DAPI staining. The migration of A549 cells was enhanced under acidic conditions and reduced by S0859 treatment (Figure 4I,J). These results suggested that S0859 partially inhibited cell motility by attenuating the activity of NBC and not by altering the protein expression of NBCn1.

### 3.5. NBCn1 Knockdown Reduced Cell Motility

To confirm the role of NBCn1 in cell motility, the cells were treated with SiRNA-NBCn1. The efficacy of SiRNA-NBCn1 was evaluated by Western blotting (Figure 5A,B). It was observed that the expression of IRBIT was independent of NBCn1 knockdown (Figure 5A,C). Membrane localization of NBCn1 decreased in the presence of SiRNA-NBCn1 (Figure 5D,E). NBCn1 knockdown dramatically inhibited the activity of NBC (Figure 5F,G). In order to verify the role of NBCn1 on the invasive motility of cancer cells, migration of the invasive A549 cells into the agarose spots was compared between the control and NBCn1 knockdown conditions. Invasive motility of the cancer cells was inhibited following NBCn1 knockdown (Figure 5H,I). The reduced migration in the presence of SiRNA-NBCn1 was also observed in other cancer cell line MCF-7, breast cancer cell line (Appendix A). These results suggest that NBCn1 is involved in cell migration. In addition, knockdown of NBCn1 or IRBIT (Appendix A) and overexpression of NBCn1 or IRBIT (Appendix A) did not affect the expression of epithelial–mesenchymal transition markers.

### 3.6. IRBIT Expression Preserved the Expression of NBCn1 in the Plasma Membrane

Epidermal growth factor (EGF)/EGF receptor (EGFR) signaling is a key factor in tumor growth and the development of metastasis, and is considered a target for chemotaxis in cancer therapy [44,45,46,47]. In order to explore the role of EGF signaling on cell migration, migration of invasive A549 cells into agarose spots was compared between EGF (recombinant human EGF)-treated and control conditions, and between IRBIT-overexpressed and IRBIT-knockdown conditions (Figure 6A). Invasive motility of the cells following EGF treatment was found to be approximately 1.4-fold higher than that of the control (Figure 6B). IRBIT overexpression had additive effects with EGF whereas IRBIT knockdown inhibited cell migration even in presence of EGF (Figure 6C). To evaluate the role of EGF in NBCn1 expression, the A549 cells were immunostained with NBCn1 in presence of EGF. Treatment with EGF revealed the enhanced expression of NBCn1 in the plasma membrane (Figure 6D). EGF treatment enhanced the expression of NBCn1 and IRBIT in the plasma membrane (Figure 6E,F). Migratory ability of the cells in the transwell assay performed using DAPI (Figure 6G,H) and crystal violet (Appendix A) was further confirmed by the results of the agarose spot assay following EGF treatment. These results indicated that IRBIT maintained NBCn1 expression and modulated EGF signaling, which is necessary for cell migration.

### 3.7. Combined Inhibition of CaMKII and NBC Markedly Inhibited Cell Migration

IRBIT functions downstream of CaMKII, and is phosphorylated by the CaMKII protein [1,6]. To evaluate therapeutic candidates that target cell migration by inhibiting the phosphorylation of IRBIT, the cells were treated with the NBC inhibitor S0859, CaMKII inhibitor KN93 [34], and a combination of both inhibitors. Migration of A549 cells was observed by DAPI (Figure 7A,B) and crystal violet staining (Appendix A), which revealed that cell migration was dramatically reduced by the combined treatment. The protein expression of NBCn1 was reduced by KN93 and combined treatment, respectively (Figure 7C,D). The protein expression of IRBIT was reduced by presence of KN93 (Figure 7C,E). To confirm the role of KN93 on NBCn1 expression, cells were treated with KN93. The inhibitory role of KN93 was revealed by a reduction in the expression of NBCn1 in the membrane and in protein expression of NBCn1 (Figure 7F,G). The expression of NBCn1 was dose-dependently reduced by the treatment of KN93 (Figure 7H,I), whereas KN93 alone did not affect the expression of IRBIT protein (Figure 7H). We also confirmed the role of CaMKII on NBCn1 with KN62 as a structural shared form of KN93. Migration was reduced in presence of KN62, whereas combined treatment with S0859 showed no additive effect on migration (Appendix A). KN62 treatment and combined treatment did not affect the expression of NBCn1 and IRBIT protein (Appendix A). These results suggest that NBCn1 is necessary for cell migration with involvement of CaMKII/IRBIT pathway.

## 4. Discussion

Taken together, this study clearly demonstrates that the IRBIT protein, which binds to NBCn1, plays a positive role in the migration of lung cancer cells by maintaining stable NBCn1 expression in the plasma membrane by mimicking the stimulatory effect of EGF. The illustrated scheme of our results is represented in Figure 8.

The role of IRBIT in tumor progression is paradoxical in that its nuclear expression has been shown to have a tumor-promoting role in chicken ovarian cancer cells but a tumor-suppressive role in human epithelial ovarian cancer cells [48]. Genomic fusion of fibroblast growth factor and IRBIT has been demonstrated in patients with cholangiocarcinoma [49]. However, IRBIT antagonizes the Bcl-2 oncogene [50] and is considered a tumor suppressor protein. Although the regulatory role of IRBIT on the expression of oncogenes and subsequent carcinogenesis is yet unknown and requires further investigation, our results demonstrate that the effect of IRBIT on the activity of ion transporters has a prominent role in cancer cell migration. It has been demonstrated that IRBIT interacts with and regulates the activity of ion transporters, such as NBCe1-B [51], CFTR [2,30], SLC26A6 [30], and NHE3 [52]. Enhanced expression and activity of NBCn1 have been demonstrated in breast cancer and the development of cancer [26,27]. Modulation of pH at the tumor microenvironment and the cancer metastasis are associated with the activity of ion transporters and channels [53,54].

The results of this study clearly demonstrate that the modulatory effect of IRBIT on the ion transporter NBCn1 enhanced cancer cell migration. IRBIT knockdown revealed that the cells had a greater tendency to migrate toward an acidic pH, compared to that of the cells in the control setup. These results demonstrate that stable NBCn1 expression was suppressed by the reduced IRBIT expression and the reduction in NBCn1 expression was associated with insufficient bicarbonate influx, which caused the cytosolic pH to become acidic, as depicted in Figure 3G. As depicted in Figure 6E,F, stimulation by EGF induced NBCn1 overexpression, but not that of IRBIT; however, recruitment of IRBIT to NBCn1 was enhanced by stimulation with EGF. It can therefore be assumed that oncogenic chemotactic signals may facilitate the expression of transporter proteins.

Moreover, the membrane expression of the enhanced transporter proteins is also maintained by the recruited IRBIT. It has been suggested that IRBIT is localized in membrane and cytosolic fraction [55]. The molecular mechanism of IRBIT on modulation of NBCn1 can be inferred by the interaction between IRBIT and NBCe1-B [5], and our previous report suggested that positively charged N-terminal domain of NBCe1-B interacts with negatively charged domain of IRBIT [2,31]. IRBIT recruits protein phosphatase 1 for activation and phosphorylation by STE20/SPS1-related kinase (SPAK) inhibits NBCe1-B. Convergent regulation occurred in the N-terminal domain of NBCe1-B. Homology between the N-terminal domains of NBCe1-B and NBCn1 predicts their similar molecular interaction with IRBIT. Thus, the role of IRBIT such as in phosphatase recruitment and kinase inhibition (e.g., SPAK) may also mediate stable membrane expression of NBCn1 [2,31].

It was notable that the effects of combined treatment with the CaMKII inhibitor KN93 and NBC inhibitor S0859 were more dramatic than those following treatment with the S0859 alone. One possibility could be that IRBIT recruits various receptors and transporters and thereby regulates a wide range of transporters, including NBCn1 [1]. Additionally, the IRBIT protein was ubiquitously detected in most of tissues such as lung, kidney, thymus, ovary, and brain, especially in the cerebrum and cerebellum [8]. Although we only elucidated the role of IRBIT on NBCn1 transporters in lung cancer cells, the potential regulatory role of IRBIT on various transporters should be focused on in other types of cancer tissue. Moreover, in order to precisely understand the modulatory effect of IRBIT on cancer migration, the dominant transporters and receptors that are necessary for cell migration need to be studied. Additionally, the molecular mechanism underlying the regulatory effect of IRBIT needs to be elucidated.

## Figures and Tables

**Figure 1 pharmaceutics-12-00816-f001:**
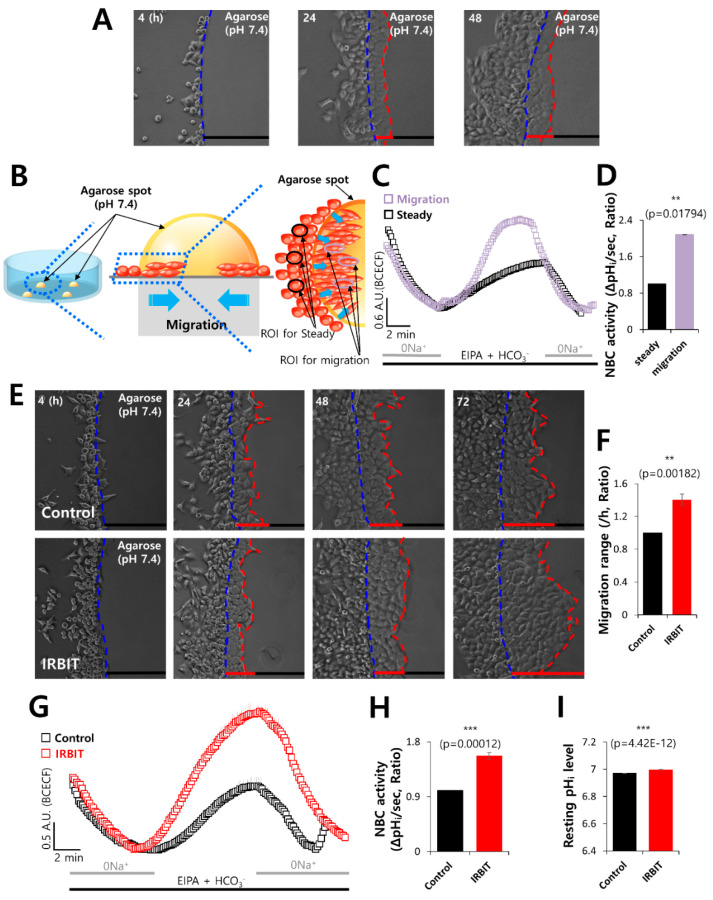
IRBIT enhanced cell motility via NBC. (**A**) Time-dependent representative images of the migration of A549 cells toward the agarose spots containing PBS (pH 7.4) for indicated times. The blue dotted lines indicate the direction of migration across the boundary of the agarose spot, which is indicated by a dashed curve. The red dotted lines indicate the lineage of cells that migrated toward the agarose spots. (**B**) Schematic illustration of agarose spots containing PBS (pH 7.4), with the red dotted lines indicating the region of interest (ROI) of migration area, and the blue dotted lines indicating ROI of steady area. (**C**) NBC activity in migration state (violet open square) and steady state (black open square) of A549 cells in the agarose spots at 48 h. Averaged traces are presented. (**D**) The bars indicate the mean ± SEM of data obtained from five experimental replicates (*n* = 5). (**E**) Time-dependent representative images of the migration of A549 cells toward the agarose spots containing PBS (pH 7.4), with or without IRBIT. (**F**) Analysis of the migration range per hour. The bars indicate the mean ± SEM of data (*n* = 10). (**G**) NBC activity in the A549 cells with (red open square) or without (control, black open square) IRBIT at 48 h. Averaged traces are presented. (**H**) The bars indicate the mean ± SEM of data (*n* = 5). (**I**) The bars depict the resting pH_i_ level of A549 cells and indicate the mean ± SEM of data (*n* = 30).

**Figure 2 pharmaceutics-12-00816-f002:**
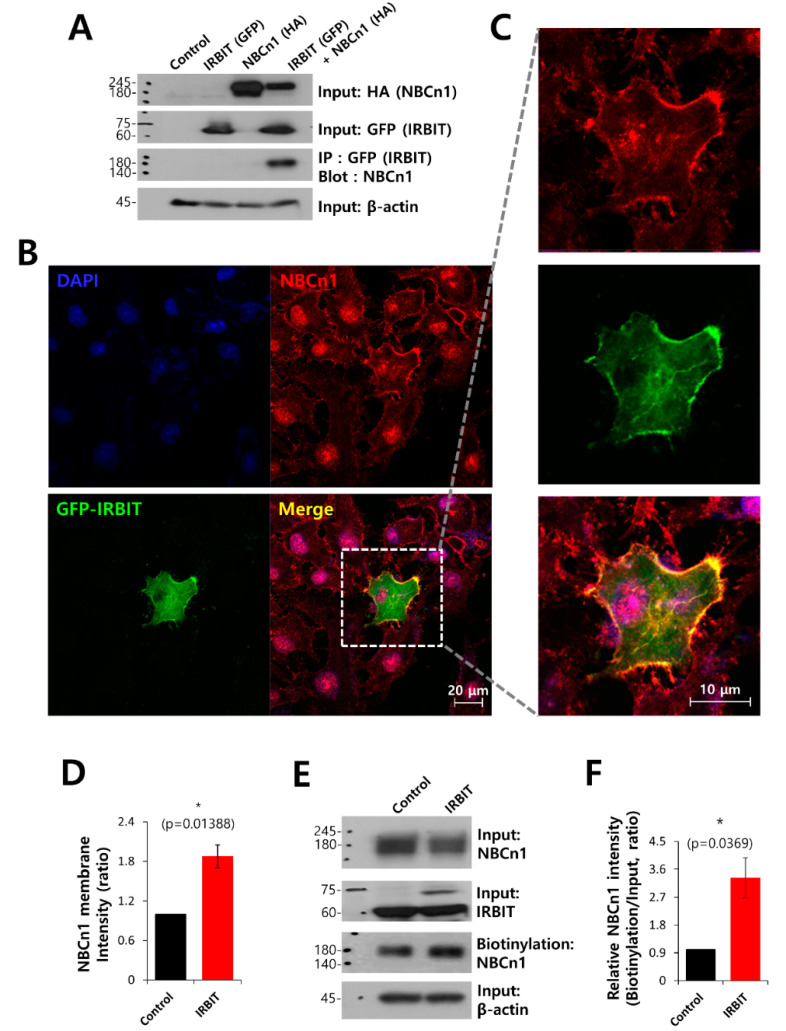
IRBIT co-localized with NBCn1 and modulated the membrane expression of NBCn1. (**A**) Co-IP of NBCn1 and IRBIT in HEK293T cells. Cells were transfected with HA-tagged NBCn1 and GFP-tagged IRBIT, immunoprecipitated with GFP antibody, and blotted with NBCn1 antibody. β-Actin, input HA, and input GFP blots were used as loading controls (*n* = 3). (**B**) Immunostaining images of NBCn1 (red), GFP-IRBIT (green), and nucleus (DAPI, blue) in A549 cells. The scale bars represent 20 μm. (**C**) Magnified images of the dotted lines. (**D**) Analysis of NBCn1 intensity in the membrane expression. The bars indicate the mean ± SEM (*n* = 5). (**E**) Surface expression of NBCn1 in the presence of IRBIT in A549 cells. Input β-actin and input NBCn1 blots were used as loading controls. (**F**) The bars indicate the mean ± SEM of relative NBCn1 intensity normalized by input NBCn1 (*n* = 3).

**Figure 3 pharmaceutics-12-00816-f003:**
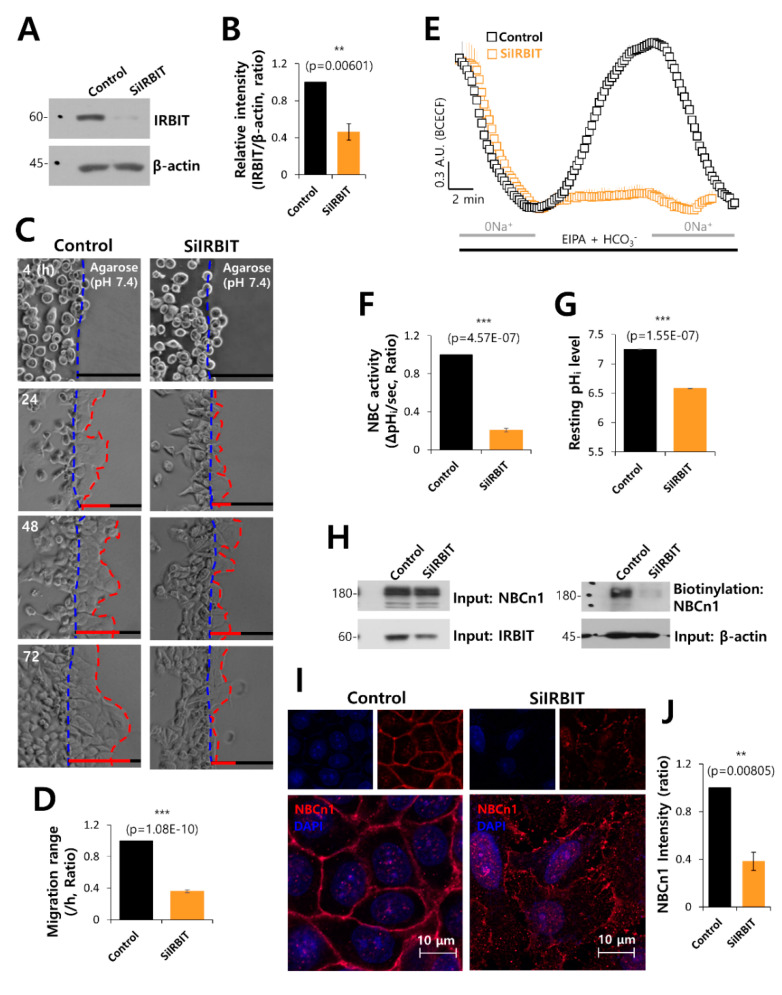
IRBIT knockdown decreased cell motility and NBC activity. (**A**) Protein expression of IRBIT in the presence of SiRNA-IRBIT (SiIRBIT) in A549 cells. β-Actin was used as loading control. (**B**) Analysis of IRBIT expression in the presence of SiIRBIT in A549 cells. The bars indicate the mean ± SEM of data (*n* = 5). (**C**) Time-dependent representative images of the migration of A549 cells toward the agarose spots containing PBS (pH 7.4), with or without SiIRBIT for indicated times. (**D**) Analysis of migration range per hour. The bars indicate the mean ± SEM of data (*n* = 6). (**E**) NBC activity in A549 cells with (orange open square) or without (control, black open square) SiIRBIT at 48 h. Averaged traces are presented. (**F**) The bars indicate the mean ± SEM of data (*n* = 4). (**G**) The bars depict the resting pHi level of A549 cells, which indicate the mean ± SEM of data (*n* = 30). (**H**) Surface expression of NBCn1 in the presence or absence of SiIRBIT in A549 cells. β-Actin and input NBCn1 blots were used as loading controls (*n* = 3). (**I**) Immunostaining images of NBCn1 (red) and nucleus (DAPI, blue) in the presence or absence of SiIRBIT. Scale bars represent 10 μm. (**J**) Analysis of NBCn1 expression in the presence of SiIRBIT in A549 cells. The bars indicate the mean ± SEM of data (*n* = 6).

**Figure 4 pharmaceutics-12-00816-f004:**
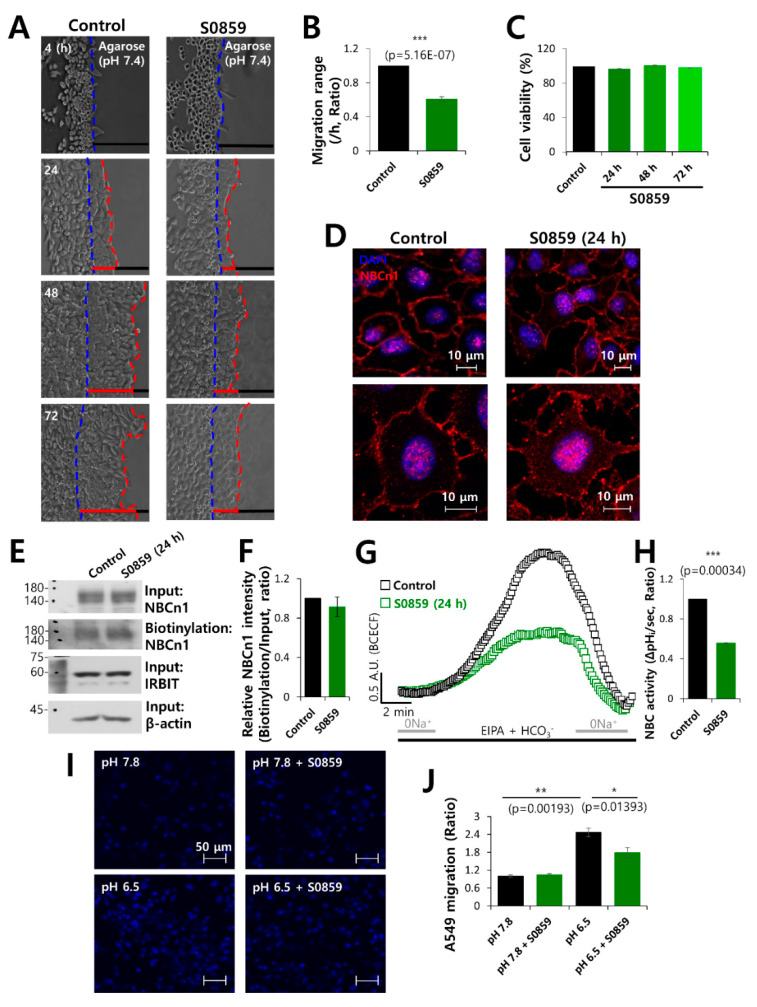
The NBC inhibitor, S0859, partially inhibited cell motility. (**A**) Time-dependent representative images of migration of A549 cells toward the agarose spots containing PBS (pH 7.4) in the presence or absence of the NBC inhibitor, S0859, at 20 μM for indicated times. (**B**) Analysis of migration range per hour. Bars indicate the mean ± SEM of data (*n* = 4). (**C**) MTT assay of A549 cells treated with 20 μM S0859 for 72 h. (**D**) Immunostaining images of NBCn1 (red) and nucleus (DAPI, blue) in the presence of S0859 at 24 h. Scale bars represent 10 μm. (**E**) Surface expression of NBCn1 24 h after treatment with 20 μM S0859. IRBIT, β-Actin, and input NBCn1 were used as loading controls. (**F**) The bars indicate the mean ± SEM of relative NBCn1 intensity normalized by input NBCn1 (*n* = 3). (**G**) NBC activity in A549 cells with (green open square) or without (control, black open square) 20 μM S0859 at 24 h. Averaged traces are presented. (**H**) Bars indicate the mean ± SEM of NBC activity (*n* = 3). (**I**) A549 migration was measured by a transwell membrane migration assay. The cells were incubated with the media and reagents indicated hereafter (bottom chamber: media of pH 7.8 and pH 6.5; upper chamber: with or without 20 μΜ S0859). Immunofluorescence staining of DAPI represents blue color. Scale bars represent 50 μm. (**J**) Analysis of the total intensity of DAPI for determining A549 cell migration. The bars represent the mean ± SEM of data (*n* = 4).

**Figure 5 pharmaceutics-12-00816-f005:**
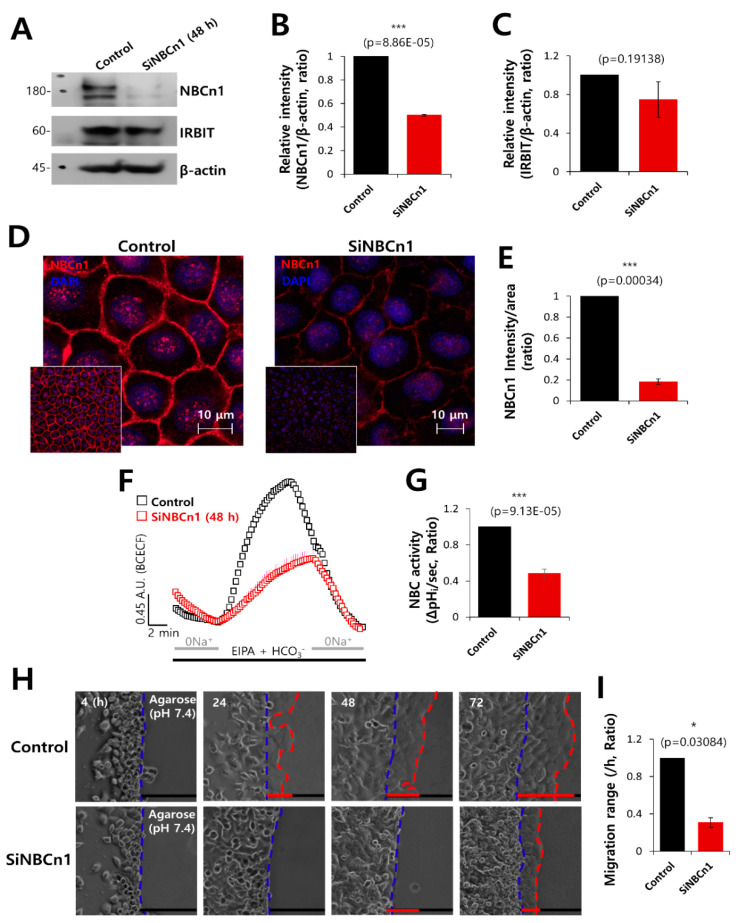
NBCn1 knockdown reduced cell motility. (**A**) Protein expression of IRBIT and NBCn1 in A549 cells in the presence of SiRNA-NBCn1 (SiNBCn1). IRBIT and β-actin were used as loading controls. (**B**, **C**) Analysis of the relative intensity of NBCn1 and IRBIT. The bars indicate the mean ± SEM of the intensities of NBCn1 and IRBIT normalized by β-Actin (*n* = 3). (**D**) Immunostaining images of NBCn1 (red) and nucleus (DAPI, blue), with or without SiNBCn1. The smaller panels represent the original magnified images. The scale bars in the magnified images represent 10 μm. (**E**) The bars indicate the mean ± SEM of the intensity of NBCn1 per area (*n* = 3). (**F**) NBC activity in A549 cells with (red open square) or without (control, black open square) SiNBCn1 at 48 h. Averaged traces are presented. (**G**) The bars indicate the mean ± SEM of NBC activity (*n* = 4). (**H**) Time-dependent representative images of migration of A549 cells toward the agarose spots containing PBS (pH 7.4) and in the presence or absence of SiNBCn1 for indicated times. (**I**) Analysis of migration range per hour. The bars indicate the mean ± SEM of data (*n* = 5).

**Figure 6 pharmaceutics-12-00816-f006:**
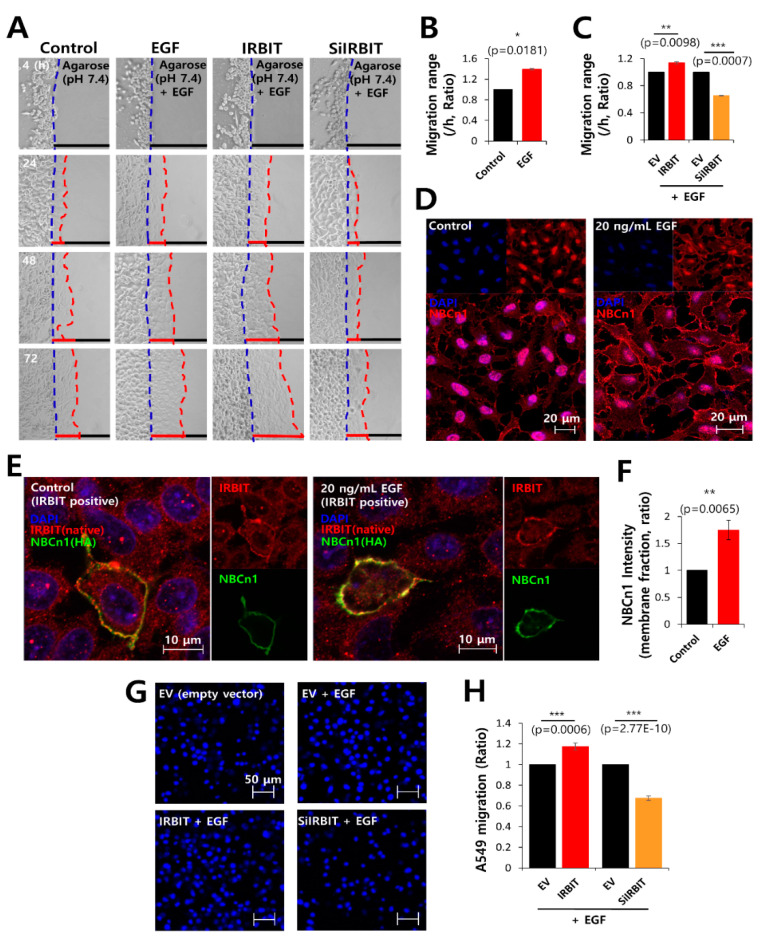
IRBIT expression preserved the expression of NBCn1 in the plasma membrane. (**A**) Time-dependent representative images of migration of A549 cells toward the agarose spots containing PBS (pH 7.4) and 20 ng/mL EGF, along with IRBIT and SiRNA-IRBIT (SiIRBIT) for indicated times. (**B**,**C**) Analysis of the migration range per hour. The bars indicate the mean ± SEM of data (*n* = 4). (**D**) Immunostaining images of NBCn1 (red) and nucleus (DAPI, blue), in the presence or absence of 20 ng/mL EGF in A549 cells. The scale bars of the magnified images represent 20 μm. (**E**) Immunostaining images of IRBIT (red), nucleus (DAPI, blue), and NBCn1 (green) in the presence or absence of 20 ng/mL EGF in A549 cells. The scale bars of the magnified images represent 10 μm. (**F**) The bars indicate the mean ± SEM of NBCn1 intensity per membrane fraction (*n* = 4). (**G**) A549 migration was measured by a transwell membrane migration assay. The cells were incubated with reagents indicated hereafter (bottom chamber: with or without 20 ng/mL EGF; upper chamber: with empty vector [EV, Control], IRBIT, and SiIRBIT). Immunofluorescence staining images with DAPI (blue). The scale bars represent 50 μm. (**H**) Analysis of the total intensity of DAPI for A549 migration. The bars present the mean ± SEM of data (*n* = 4).

**Figure 7 pharmaceutics-12-00816-f007:**
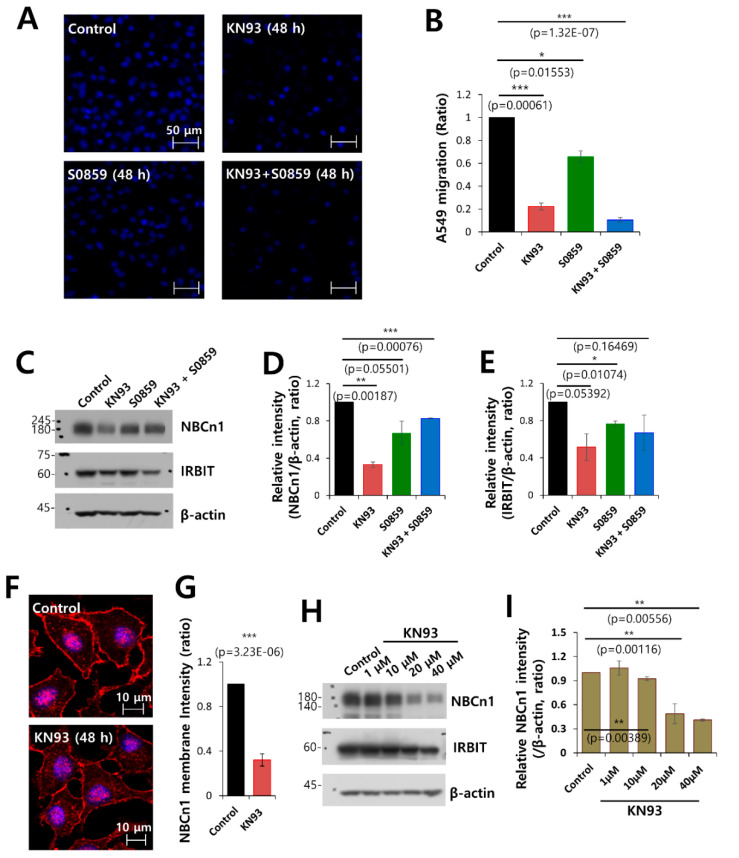
Combined inhibition of CaMKII and NBC markedly inhibited cell migration. (**A**) Migration of A549 cells was measured by a transwell membrane migration assay. The cells were incubated with the medium and reagents indicated hereafter (bottom chamber: media of pH 6.5; upper chamber: pretreatment with KN93 and S0859 at 48 h). Immunofluorescence staining with DAPI (blue). The scale bars represent 50 μm. (**B**) Analysis of the total intensity of DAPI for determination of A549 migration. The bars represent the mean ± SEM of data (*n* = 4). (**C**) Protein expression of IRBIT and NBCn1 in A549 cells in the presence of KN93 and S0859 at 20 μM. β-Actin was used as a loading control. (**D**) Analysis of the relative intensity of NBCn1. The bars indicate the mean ± SEM of the intensities of NBCn1 normalized by β-Actin (*n* = 3). (**E**) Analysis of the relative intensity of IRBIT. The bars indicate the mean ± SEM of the intensities of IRBIT normalized by β-Actin (*n* = 3). (**F**) Immunostaining images of NBCn1 (red) and nucleus (DAPI, blue) in the presence or absence of 20 μM KN93. Scale bars of the magnified images represent 10 μm. (**G**) The bars indicate the mean ± SEM of the NBCn1 membrane determined from three experimental replicates (*n* = 3). (**H**) Protein expression of IRBIT and NBCn1 in A549 cells in the presence of KN93 at 1–40 μM. β-Actin was used as a loading control. (**I**) Analysis of the relative intensity of NBCn1. The bars indicate the mean ± SEM of the intensities of NBCn1 normalized by β-Actin (*n* = 3).

**Figure 8 pharmaceutics-12-00816-f008:**
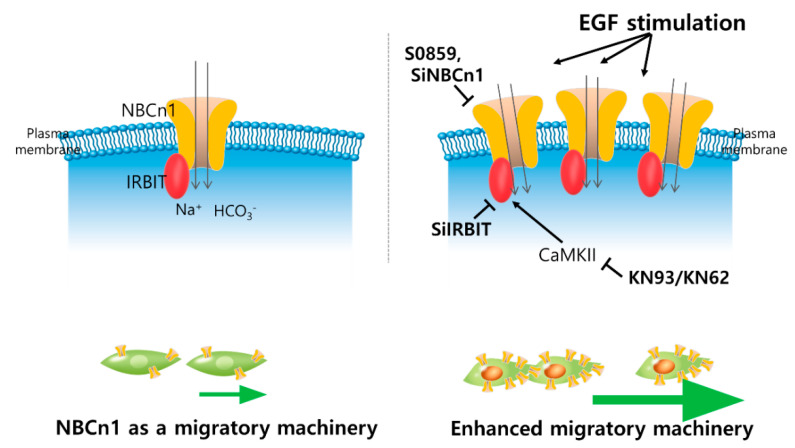
Schematic illustration of function of NBCn1 as a migration machinery and regulatory factors on NBCn1-mediated migration in lung cancer cells. IRBIT protein is recruited to NBCn1, plays a positive role in the migration of lung cancer cells and stored expression of NBCn1 in the plasma membrane as a stimulated condition with EGF. NBC inhibitor S0859, knockdown of NBCn1 or IRBIT, CaMKII inhibitor KN93 attenuate the NBCn1-mediated cellular migration.

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
