# Peer review of "Protective Role of IRBIT on Sodium Bicarbonate Cotransporter-n1 for Migratory Cancer Cells"

_pharmaceutics, 2020, doi:10.3390/pharmaceutics12090816_

Round 1
Reviewer 1 Report
The manuscript is interesting and well written. It requires some suggestions to improve the quality of this study.
1. Authors claimed that Knockdown of IRBIT or NBCn1, and treatment with an NBC-specific inhibitor, S0859, attenuated cell migration. According to my opinion please perform cell migration and invasion assay on the Boyden chamber. I would recommend to authors that please take an idea from the latest papers-Parashar D et al - Cell Reports 2019 (PMID: 31875548), Chen et al 2020,- Oncogene (PMID: 32029900). Where the authors showed migration and invasion assays.
2. It will be appreciable if authors could provide some key experiments related to EMT markers by western blot or immunofluorescence technique.
3. Schema is required with a complete representation of the hypothesis.
Author Response
Dear reviewer and editor,
Before addressing each of the comments below, we appreciate the reviewers for the valuable comments and careful consideration. We obviously have needed to quote all sources correctly and done so at the places where we had missed before. In addition, the manuscript has been edited to make appropriate information and additional data to this body of work.
Responses to comments of reviewers as below:
Reviewer 1.
The manuscript is interesting and well written. It requires some suggestions to improve the quality of this study.
- Authors claimed that Knockdown of IRBIT or NBCn1, and treatment with an NBC-specific inhibitor, S0859, attenuated cell migration. According to my opinion please perform cell migration and invasion assay on the Boyden chamber. I would recommend to authors that please take an idea from the latest papers-Parashar D et al - Cell Reports 2019 (PMID: 31875548), Chen et al 2020,- Oncogene (PMID: 32029900). Where the authors showed migration and invasion assays.
-response: We appreciate your comment and yes, we already performed migration assay on transwell Boyden chamber and also performed conformational experiments with crystal violet analysis and scratch wound healing assay in supplementary figures. We considered the migratory ability with DAPI and crystal violet assay and we also used Boyden chamber within 6 hrs incubation. In addition, we verified the migration assay between DAPI and crystal violet staining in presence of IRBIT plasmids in Supplementary Figure 1. Technically, Parashar D et al proposed and performed experiments at 37 °C for 12 h for migration assays and for 16h for invasion assays. Thanks for recommendation. We will consider invasion assay for future experiment.
- It will be appreciable if authors could provide some key experiments related to EMT markers by western blot or immunofluorescence technique.
-response: We appreciate your comment. We performed the expression of EMT markers such as N-cadherin, E-cadherin, and vimentin in various experimental conditions such as knockdown of IRBIT and NBCn1 and overexpression of IRBIT, NBCn1, and IRBIT+NBCn1. We found that there were no changes of EMT marker expression and added the additional data in Supplementary Figure 5 and 6.
- Schema is required with a complete representation of the hypothesis.
-response: We appreciate your comment. We added schema to present proposed mechanism in Figure 8.
Reviewer 2 Report
The paper by Hwang et. al shows that the IRBIT protein, which binds to NBCn1, plays a positive role in the migration of lung cancer cells by maintaining stable NBCn1 expression at the plasma membrane, by mimicking the stimulatory effect of EGF. Inhibition of both IRBIT and NBCn1 inhibited the migration of cancer cells. The paper is well written, and the results are clearly presented.
Author Response
Dear reviewer and editor,
Before addressing each of the comments below, we appreciate the reviewers for the valuable comments and careful consideration. We obviously have needed to quote all sources correctly and done so at the places where we had missed before. In addition, the manuscript has been edited to make appropriate information and additional data to this body of work.
Responses to comments of reviewers as below:
Reviewer 2.
The paper by Hwang et. al shows that the IRBIT protein, which binds to NBCn1, plays a positive role in the migration of lung cancer cells by maintaining stable NBCn1 expression at the plasma membrane, by mimicking the stimulatory effect of EGF. Inhibition of both IRBIT and NBCn1 inhibited the migration of cancer cells. The paper is well written, and the results are clearly presented.
-response: We appreciate your favorable consideration.
Reviewer 3 Report
Comments to Author:
In the manuscript “Protective role of IRBIT on sodium bicarbonate 3 cotransporter-n1 for migratory cancer cells”, the authors demonstrated that NBCn1 activity through IRBIT regulate cancer cell migration. This manuscript may contribute to the knowledge in the field of cancer. However, current version of this manuscript does not provide the level of depth and the impact required for the journal “Pharmaceutics”. I think that the pharmaceutical demonstration of combined modulation of sodium bicarbonate 3 cotransporter-n1 and IRBIT still have some problems as indicated below.
Major comments
- Authors discuss “It was noteworthy that the effects of combined treatment with the CaMKII inhibitor KN93 and NBC inhibitor S0859 were more dramatic than those following treatment with the S0859 alone” in discussion section. However, In Figure 7D and E, combined treatments are not dramatic changes regarding NBCn1 and IRBIT. These data are weak evidences to support the suppression of A549 migration. Although authors describe in Results section “The protein expression of IRBIT was reduced by combined treatment in Figure 7E”, treatment with only KN93 is most reduced than combined treatment. It is wrong to interpret experimental results based on only p value.
- In Figure 7D and 7G, the protein expression level of NBCn1 is inconsistent with the membrane localization level of NBCn1 when cells were treated with KN93. Authors should add examinations using some concentrations of KN93 or other CaMKII inhibitors.
- In Figure 7C,D and 7H,I, authors repeated to confirm the role of KN93. Authors should perform examinations using other CaMKII inhibitors to confirm IRBIT function downstream.
- Authors write “migratory cancer cells” in title. But only one cell line was used in this study. More cell lines are recommended.
- Authors fail to explain the methods of statistic analysis including t-test, ANOVA, and so on. Authors should write statistical methods and software.
- IRBIT is located in endoplasmic reticulum. Authors should explain the cellular kinetics of IRBIT including molecular mechanism of IRBIT on modulation of NBCn1 in migratory cancer cells.
- Introduction, authors should write the information regarding role of NBCn1 and IRBIT in clinical cancer because clinical information is important in order to develop therapeutic strategy based on this study in future.
Author Response
Dear reviewer and editor,
Before addressing each of the comments below, we appreciate the reviewers for the valuable comments and careful consideration. We obviously have needed to quote all sources correctly and done so at the places where we had missed before. In addition, the manuscript has been edited to make appropriate information and additional data to this body of work.
Responses to comments of reviewer as below:
---
Comments to Author:
In the manuscript “Protective role of IRBIT on sodium bicarbonate 3 cotransporter-n1 for migratory cancer cells”, the authors demonstrated that NBCn1 activity through IRBIT regulate cancer cell migration. This manuscript may contribute to the knowledge in the field of cancer. However, current version of this manuscript does not provide the level of depth and the impact required for the journal “Pharmaceutics”. I think that the pharmaceutical demonstration of combined modulation of sodium bicarbonate 3 cotransporter-n1 and IRBIT still have some problems as indicated below.
Major comments
1. Authors discuss “It was noteworthy that the effects of combined treatment with the CaMKII inhibitor KN93 and NBC inhibitor S0859 were more dramatic than those following treatment with the S0859 alone” in discussion section. However, In Figure 7D and E, combined treatments are not dramatic changes regarding NBCn1 and IRBIT. These data are weak evidences to support the suppression of A549 migration. Although authors describe in Results section “The protein expression of IRBIT was reduced by combined treatment in Figure 7E”, treatment with only KN93 is most reduced than combined treatment. It is wrong to interpret experimental results based on only p
-response: We appreciate your valuable comment. Use of CaMKii inhibitor is provided the inhibition of modulatory role of IRBIT on NBCn1. We re-analyzed the western blot data and performed additional experiment of KN-93 at various dose in Figure 7H and 7I. As you recommended, we rephrased experimental results.
2. In Figure 7D and 7G, the protein expression level of NBCn1 is inconsistent with the membrane localization level of NBCn1 when cells were treated with KN93. Authors should add examinations using some concentrations of KN93 or other CaMKII inhibitors.
-response: We appreciate your comment. We combine the answer for comment 2 and 3.
3. In Figure 7C,D and 7H,I, authors repeated to confirm the role of KN93. Authors should perform examinations using other CaMKII inhibitors to confirm IRBIT function downstream.
-response: We appreciate your comment. As an answer for comment 2 and 3, we performed experiments with another CaMKii inhibitor KN62. Actually, KN-93 (Front Pharmacol. 2014; 5: 21) is the most widely used inhibitor for study of cellular and in vivo functions of CaMKII. We urgently got KN-62 to perform the experiment. KN-62 shares structural elements and mechanism of action with KN-93. It is known that KN-62 interferes with the ability of Ca2+/CaM. The treatment of KN-62 also reduced migration as shown in Supplementary Figure 9, whereas did not affect the expression of NBCn1. We understand your point, however, in this study we used CaMKII inhibitor to block the role of IRBIT. We also understand that we cannot rule out the effect of CaMKII inhibitor on NBCn1. Please consider our main focus on the regulatory role of IRBIT on NBCn1 from the perspective of cellular migration in this article.
4. Authors write “migratory cancer cells” in title. But only one cell line was used in this study. More cell lines are recommended.
-response: We appreciate your comment. We performed additional migration experiment with breast cancer cell line MCF-7 cells which express NBCn1. As shown in Supplementary Figure 4, the treatment of siRNA-NBCn1 reduced the migration of MCF-7 suggesting that NBCn1 also can be a migratory machinery in MCF-7 cells.
5. Authors fail to explain the methods of statistic analysis including t-test, ANOVA, and so on. Authors should write statistical methods and software.
-response: We appreciate your comment and the methods has been edited to make appropriate information.
6. IRBIT is located in endoplasmic reticulum. Authors should explain the cellular kinetics of IRBIT including molecular mechanism of IRBIT on modulation of NBCn1 in migratory cancer cells.
-response: We appreciate your comment. IRBIT is located in membrane fraction and cytosolic fraction including ER (Ando H et al., J. Neurochem., 109 (2009), pp. 539-550). We previously have been reported that the cellular regulatory role of IRBIT on NBCs (Proc Natl Acad Sci U S A. 2013 Mar 5; 110(10): 4105–4110.). IRBIT binds to N terminus of NBCs. Briefly, IRBIT has protein phosphatase 1 (PP1)-binding domain and recruits PP1. The PP1-recruited IRBIT restore the membrane expression of NBCe1-B to inhibit the phosphorylation of SPAK. The effect of SPAK on NBC induces the reduced expression of NBCe1-B in plasma membrane. Convergent regulation of IRBIT and SPAK is occurred in N-terminal domain. Although all evidences were from NBCe1-B, the NBCn1 and NBCe1-B have homology in the N-terminal domain and thus the cellular mechanism might be shared. Accordingly, in the current study, we performed the regulatory role of IRBIT on NBCn1, which is expressed in breast and lung cancer cell lines with view of pathological aspect.
7. Introduction, authors should write the information regarding role of NBCn1 and IRBIT in clinical cancer because clinical information is important in order to develop therapeutic strategy based on this study in future.
-response: We appreciate your valuable comment. The IRBIT protein was ubiquitously detected in most of tissues. However, the information of role of IRBIT on tumor cells is relatively low. Thus, we think there are lots of potential in cancer biology with scope of transporter physiology and involvement of regulatory protein such as IRBIT. We added this point in introduction section
Round 2
Reviewer 3 Report
The authors have answered all my questions, and the current manuscript improved.